# A mechatronic shape display based on auxetic materials

Anthony Steed[1,2], Eyal Ofek[1], Mike Sinclair[1] & Mar Gonzalez-Franco [1✉]

Shape displays enable people to touch simulated surfaces. A common architecture of such devices uses a mechatronic pin-matrix. Besides their complexity and high cost, these matrix displays suffer from sharp edges due to the discreet representation which reduces their ability to render a large continuous surface when sliding the hand. We propose using an engineered auxetic material actuated by a smaller number of motors. The material bends in multiple directions, feeling smooth and rigid to touch. A prototype implementation uses nine actuators on a 220 mm square section of material. It can display a range of surface curvatures under the palm of a user without aliased edges. In this work we use an auxetic skeleton to provide rigidity on a soft material and demonstrate the potential of this class of surface through user experiments.

[1] Microsoft Research, One Microsoft Way, Redmond, WA, USA. [2] Department of Computer Science, University College London, London, UK. ✉email: margon@microsoft.com

Shape-changing devices are a subset of robotic systems that attempt to create shapes and surfaces[1]. These can be then used to deliver haptics for humans that can encounter, touch and manipulate these shapes[2]. In contrast to articulated robots that are primarily designed to control one point of manipulation, shape displays can present a larger area and multiple possible shapes covering the whole hand. In that regard, shape-changing devices are primarily concerned with the expression of more general forms. Thus, a shape-changing device might have a reasonably high number of degrees of freedom and a mechanical structure that converts actuation to some form of constrained surface. Such systems are increasingly being investigated as both input and output devices for human-computer interfaces[3,4] and might one day become a common peripheral for our PCs or virtual reality systems.

The most typical type of shape-changing device is a pin-array (see examples in Fig. 1), where rods are actuated in height to form an approximation of a 3D surface[5]. Following Hirota & Hirose, several systems have extended the concept of pin arrays. Most displays have supported full-hand interaction creating a denser array of pins[6–8]. Others have focused on small arrays that represent small surface features under the thumb or fingers[9–11] (see Fig. 1a, b). Swarm robots can be used also for encounter haptics as discrete shape displays[12]. A second approach to display a surface is to construct an articulated surface with hinges[13–15] (see Fig. 1d). However, to scale this up a fixed pattern of hinges is needed[16–18] for flexibility in the structure[19]. A third approach is to actuate the surface by in-surface actuators, so the surface itself deforms by stretching or deforming[20–22] (see Fig. 1e, f).

The main limiting factor of these pin and hinge approaches is that they represent shape through a discrete set of rigid elements and thus only convey gross shape. When the user encounters the device, they can suffer an Uncanny Valley of Haptics[23], triggered by the relief between pins or hinging or spacing between elements. To create the impression of a smoother surface, an elastic surface can be stretched over the elements[24], but this doesn't change the frequency of the features that can be perceived

through tactile exploration, and as the surface is elastic, it is not uniform stiffness. Further, completely elastic displays also cannot render the stiffness necessary to explore hard shapes[2]. Therefore, the problem of shape displays cannot be solved with soft robotics alone, as the structure needs a skeleton that resists locally applied forces. Other relevant work comes from building techniques through molds, which are for example used to create shells of concrete panels for large curve fabrication[25]. Those skeletons provide smooth interpolations of shapes and curves, however they are not actuated and thus rely on manual reconfiguration.

The challenge is to create a pin or hinge system with an interpolating material that may bend freely to generate shapes when we actuate it. The interpolating material should retain its shape under some local pressure (as generated by touch), but be flexible over large scales.

The majority of materials become thinner as we apply a stretching force in the perpendicular direction. However, auxetic materials either retain their width or become thicker[26,27]. In material science, this is described by the material having a negative or zero Poisson's ratio[28]. While being a rarity in nature, auxetic materials created through structural design have been engineered for a variety of purposes from shoe soles to space travel accessories[26] (Fig. 2).

Auxetic surfaces can be designed to curve in two directions, so they are a promising route for the design of non-developable surfaces[29]. Auxetic surfaces can not only be shaped in two directions controlled by appropriate sets of mechatronic actuators, but they can also provide the necessary stiffness by the material, thickness, and pattern of cutting off the surface. The surface can then be coated with a skin that smooths out sharp pattern edges exposed while curving the surface (Fig. 2). With this surface, the user placing their fingers or whole hand on the surface should perceive single and double curvatures of various types. Further, the surface presents both displacement and surface normal. The latter is not conveyed accurately by the previous surface displays due to the discretization of the surface. However,

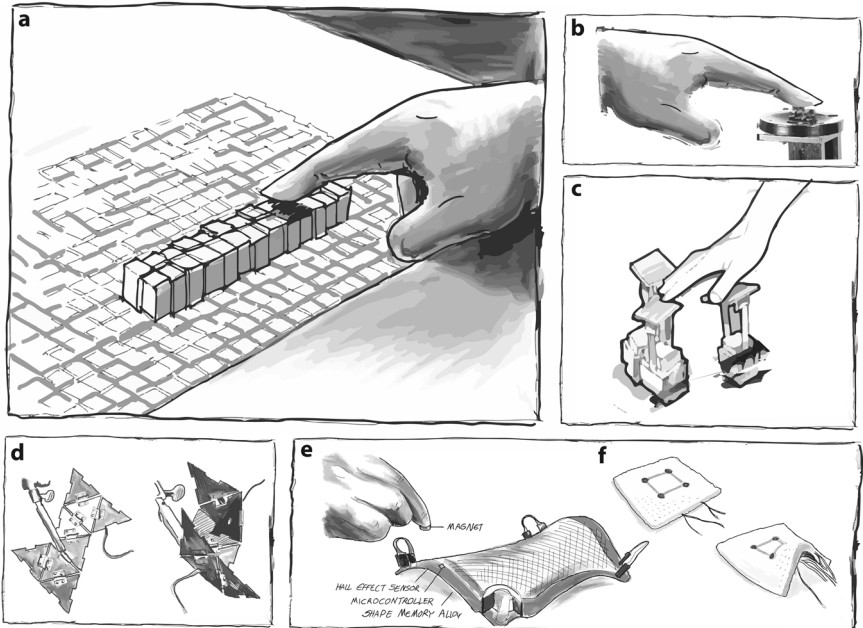

**Fig. 1 Previous shape displays. a** shape displays using pin arrays like the Inform prototype[6–8]. **b** pins can be miniaturized to fit fingertips like in the NormalTouch prototype[9,10]. **c** swarm robots can be used also for encounter haptics as sparse shape displays like in the Hapticbots prototype[12]. **d** shape displays using hinged platforms like in Robotic Origami[13–15]. **e, f** deformable shape displays made of elastic materials like NURBSforms or Surflex prototypes[20–22].

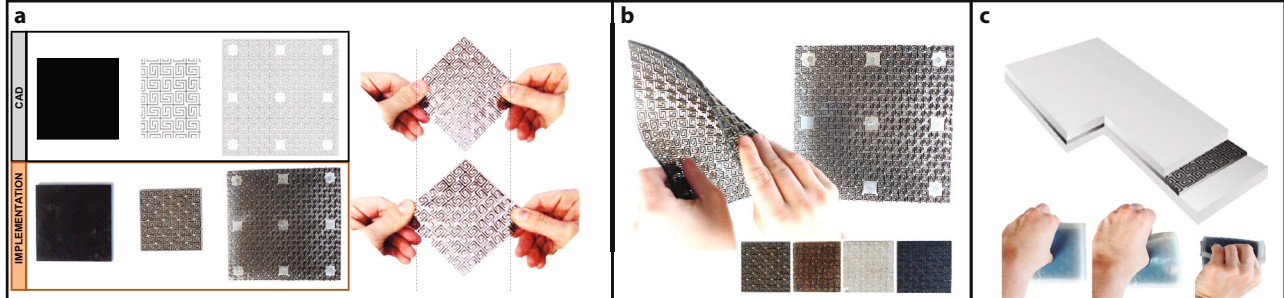

**Fig. 2 Building the auxetic surface. a** The pattern chosen was a repeating spiral pattern[45], the first of a recently explored class of Archimedean spirals that can be used to give double curvature to a flat sheet[46]. Compared to other patterns, this pattern fills the majority of the surface area and provides good support to the surface. Each spiral occupied 1 cm². **b** Experimentation with a variety of materials including wood, polycarbonate, and acrylics of different thicknesses. Finally, 6 mm sheets of polycarbonate were chosen as the material as it was extremely tough when cut with the pattern. As laser cutting proved to generate burnt crusts along with cuts, we found water cut faces to be flatter, despite small irregularities of the water jet cutter's kerf. **c** Smooth-On's Dragon Skin™ 30 silicone rubber (Shore hardness 30 A) is coated to both the front and back of the surface by a two-step process: First, molding the top surface, letting it cure sufficiently to hold the polycarbonate layer, then molding the back layer.

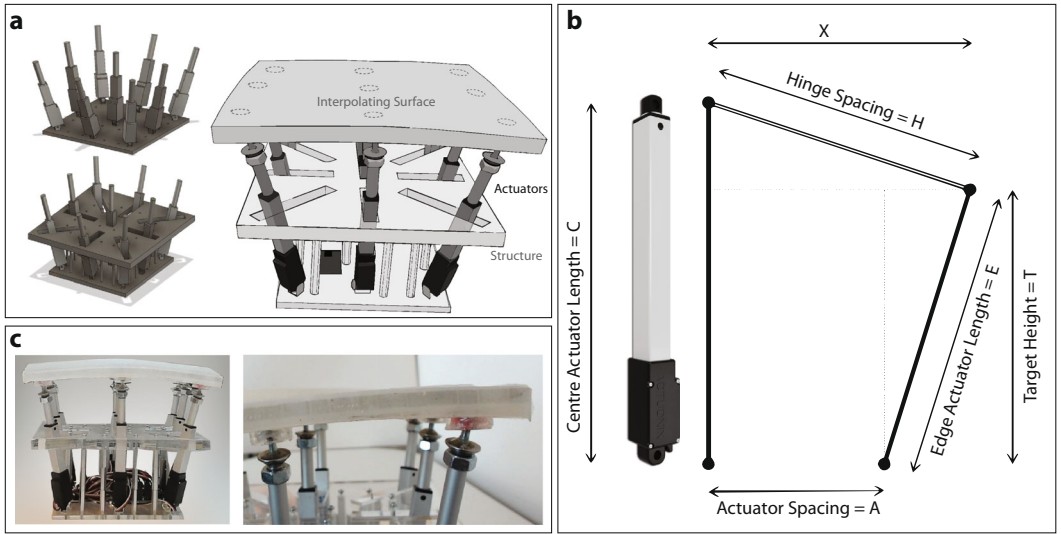

**Fig. 3 Building the shape display. a** The shape display is composed of 9 actuators mounted in a rigid assembly. The whole surface has mounting points for each actuator. The interpolating surface on the top is made from layers of acrylic and silicone rubber. The acrylic is cut with a pattern that gives it auxetic behavior. **b** The diagram of the measurements for actuator control. **c** The prototype's surface is 220 mm square. Actuators are mounted to the interpolating surface with ball joints.

surface normal is an important determinant of shape perception, dominating displacement cues in some situations[30].

We propose a shape display that is based on a flexible and stretchable auxetic material. Thanks to this surface material our shape display provide stiffness locally and enough flexibility to interpolate between larger spaces. So, with relatively few actuators we can build a mechatronic shape display that can achieve various shapes but still supports the whole hand.

The surface itself is comprised of two main components. The first is the auxetic structural material, which allows the surface to be flexible in two directions; this is the basis to interpolate the surface between the actuators. These endpoint actuators are the second component. The size of the surface was set to 22 × 22 cm so that a whole palm could be supported by the surface. The spacing between the mechanical fixtures of the actuators to the surface, 9 cm, left the edges of the surface unsupported, but the material stiffness maintained the shape of the surface (Fig. 3).

In essence, this auxetic material is a merger between soft robotics and the skeleton surface approach.

Here, we show how using auxetic materials for a shape display we can create a surface that is smooth to the touch and that

provides a range of different Gaussian curvatures (i.e. bends in two directions independently to give both positive and negative Gaussian curvatures) but is not easily pliable under forces normal to the surface (i.e. not simply elastic). The proposed display can render different curvatures, while maintaining a coverage footprint, which may enable the easy extension, tiling, and continuous rendering as the display is moved (Fig. 4).

## Results

**Stress test**. We carried a series of stress tests to measure how much flexibility the auxetic structure created in the originally rigid surfaces with and without the silicon coating (Fig. 5). We observe a high elasticity on the auxetic material that follows Hooke's law type of behavior. The measurements were taken in the furthest point from the actuators, i.e. right in the middle between them, which were situated at ~9 cm (in X and Y). Note we did not push the stress test further to avoid damaging the surfaces outside of the elastic region. Nevertheless, we observe that the final shape display can sustain forces over 50 N, with deformation below 1 cm. This performance covers most human

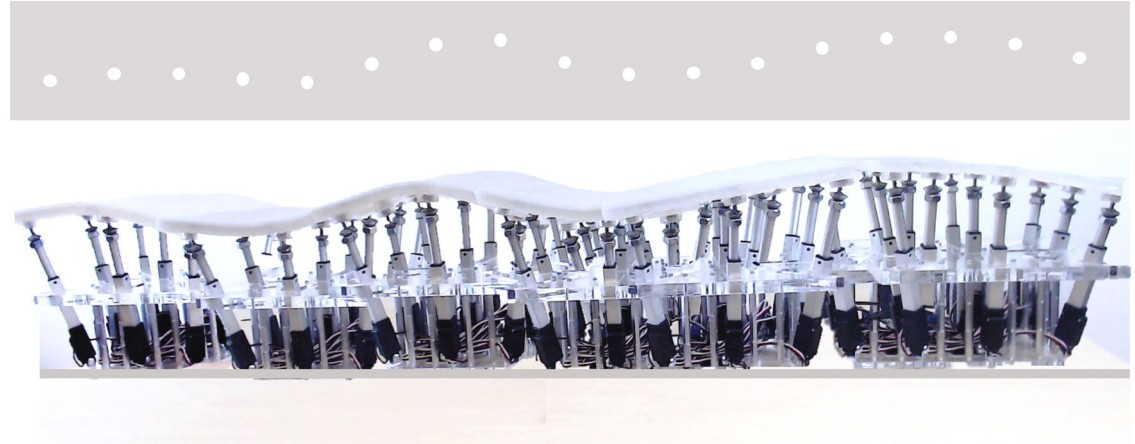

**Fig. 4 Movable shape display.** Trace of the prototype mounted on a mobile platform that allows a person to feel a shape as it moves across space.

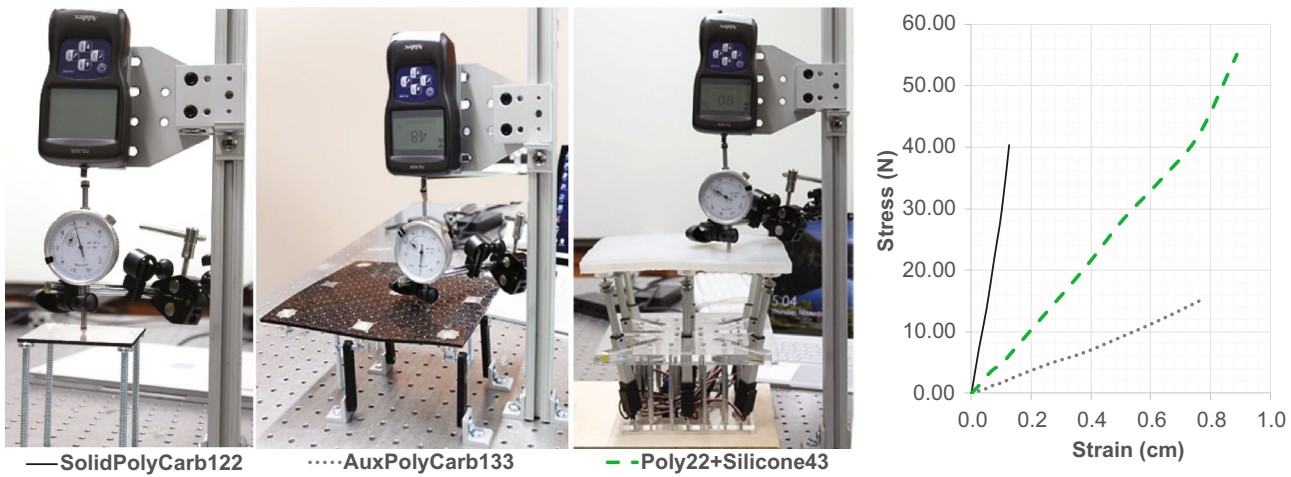

———SolidPolyCarb122　　⋯⋯⋯AuxPolyCarb133　　– –Poly22+Silicone43

**Fig. 5 Stress test.** Stress test is performed on the different materials and auxetic structures. Solid Polycarbonate of 3.1mm (0.122") thick, (SolidPolyCarb122, represented with the solid black line on the graph). A 3.34mm (0.133") thick polycarbonate with the auxetic structural cut, without coating (AuxPolyCarb133, represented with the gray dotted line in the graph). The final surface of the device with the 5.5mm (0.22") thick polycarbonate and the silicone coating, 10.1mm (0.43") total thickness (Poly22-Silicone43, represented with the green dashed line on the graph).

finger scale forces, generally humans achieve around 40 N of force when finger pressing with a neutral wrist[31].

**Experiment 1**. To validate the ability of the shape display to render different types of objects with small variations in curvature we ran a study in which participants ($n = 17$) had to choose which of four shapes they were touching: a convex spherical surface, a concave spherical surface, a convex cylindrical surface (1D curve) and a concave cylindrical surface (Fig. 6). We also tested for three different maximum height levels of (25, 10, and 5 mm) (corresponds to curvatures of the radius of 254.5, 610, and 1212 mm) up to a point in which the shape display was almost flat (5 mm). Surface orientation is a key feature in curvature perception[30].

Participants completed 36 comparison trials, each shape (x4) against all the others (x3) in the different curvatures (x3). In every trial, participants had to select which of the four possible shapes was the one they touched.

The results (Fig. 6) showed that participants had an overall accuracy of 80% in correct recognition of the shape when the curved surface-displayed has depth differences of over 25 mm, and the accuracy of recognition dropped to 42% when the surface curvature generated height differences of 5 mm. At this height, the surface looks almost flat to the user (Fig. 6). A paired

Friedman analysis showed that not all shapes were equally recognized ($\chi^2 = 13.6$, df = 3, $p = 0.033$). The surface that was most recognizable was the Convex Hill, with a 75% accuracy across all curvatures, and reaching 80% accuracy for curvatures over 10 mm. Convex Hill shapes were significantly more recognizable than the two concave shapes (Conover test with Bonferroni adjustment $p < 0.001$). No significant difference in recognition was found between the Convex Hill and the Convex Cylinder ($p = 0.09$) that also showed high accuracy scores 66%, being maximal with curvatures of 25 mm at 92% recognition (Fig. 6). These results validate the ability of the shape display as a proxy for the haptic perception of shapes of different curvatures, particularly if the shapes are presenting convexity.

Note that we believe that worse results with concavity do not lie with the display, but rather on the fact that we asked participants to lay their full palm on the display, and in that position, the hand may hover about but not touch the cavity (supported by the ridges around it) and thus limit the recognition of the shape.

**Experiment 2**. In experiment 2, we further explored the ability of participants to discriminate between surface curvatures within one type of shape (Fig. 6). As the surface is smooth, without aliasing caused by different adjacent physical sections, we would

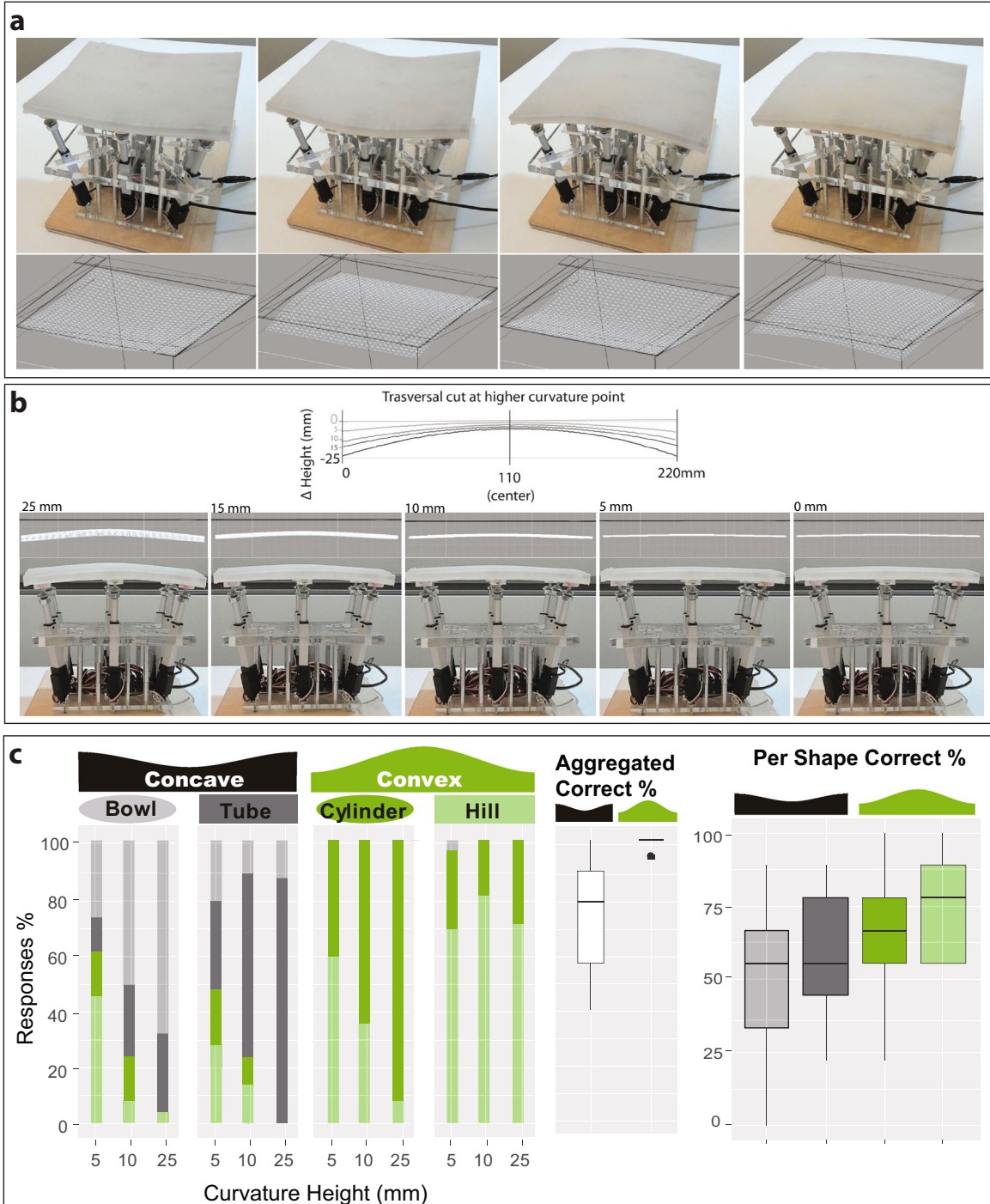

**Fig. 6 Four shapes and heights tested. a** Rendering at 25 mm of a concave bowl, concave cylinder, convex tube, and a convex hill. **b** Different heights tested. Note that the 5 mm curvature is almost flat. In the graph, the curvatures are shown as a quadratic function of the meshes measured by the Kinect depth sensor. The Y dislocation is shown only to help visualize the curvatures that go from less to more hilly. The maximum depth height (−25 mm) was a known good, and 0 mm was flat. **c** Classification responses of participants ($n = 17$ with 36 comparison trials) when presented with a particular shape in different curvatures. The boxplot on the right shows the overall ability of the shape display to render concave or convex shapes. For the boxplot: the center line represents the median; box limits represent upper and lower quartiles; whiskers the 1.5× interquartile range; and individual points are outliers.

expect the surface to be able to show very small changes in curvature.

Participants experienced a forced paired test inside virtual reality, in which they had to choose which of two shapes was more curved. We chose the Convex Hill as the main shape, and then changed the curvatures. Differences across the display on the experiment were 5.5 mm (equivalent to curvature of 1102.75 mm), and the most dramatic height difference was 25 mm (equivalent to curvature of 254.5 mm). The Convex Hill was the most accurately detected shape in Experiment 1. If the response to the paired test was correct, we decreased the height of the curvature for the next comparison, until a point where a threshold of perception was reached, and no more differences could be identified between the shapes.

**Table 1 Curvature Radius.**

| Height at the highest point | Curvature radius |
|---|---|
| 25 mm | 254.5 mm |
| 20 mm | 312.5 mm |
| 15 mm | 410.83 mm |
| 10 mm | 610 mm |
| 5 mm | 1212.5 mm |
| 0 mm | Infinite |

Conversion between maximum depth differences and the radius of curvature.

In total, there were 16 levels increasing in difficulty (see Fig. 6 for a visualization of a selection of levels and Table 1 for their radius of curvature). Curvature levels were presented on this order: 25 mm, 20 mm, 15 mm, 13 mm, 10 mm, 9 mm, 8 mm, 7 mm, 6 mm, 5.5 mm, 5.3 mm, 5.2 mm, 5.15 mm, 5.1 mm, 5.05 mm, 5 mm, and some of the equivalent curvature rations can be seen in Table 1.

At each iteration of the paired comparisons, one of the curvatures was always at the minimum height (5 mm, and 1212.5 mm radius) versus one with a higher curvature. The assignment of which curve was presented first or second was randomized. If participants chose correctly the shape with greater curvature, the experiment diminished the curve level for the forthcoming paired comparison. When participants made a mistake, the experiment moved 3 levels up in curvature. Participants could make a total of 3 mistakes. The levels at which participants made a mistake were considered the perceptual thresholds of the participant.

We measured the shape display curvatures externally. A Kinect with a depth sensor was used to scan the shape display's shape, see results in Fig. 6, and a quadric curve was fitted to the empirical points retrieved. This calibration was undertaken for each of the shapes used in both experiments.

Special care is given to the presentation of a completely flat display to avoid bias in the assessment. In prior pilots, we found that when comparing curves to a flat surface, participants would change the task and intrinsically respond to the question of: is this flat? rather than to the question: which one is more curved?

**Control condition.** In order to ensure that participants were not able to establish the curvature/shape based solely on visual clues rendered in the virtual reality, we ran a control condition combined with Experiment 2.

In this condition, participants were not allowed to touch the surface, and they were requested to perform the assessment only through visual cues. The virtual reality setup and rendering of the shape display were designed through several iterations and pilots to visually obfuscate as much as possible the curvature of the shape, therefore the assessment during the task would need to be based on tactile input rather than visual.

Results show that the mean threshold reached by each participant in the 3 tries they had (Fig. 7). Given the non-parametric nature of the paired test, we ran a paired Friedman analysis. We find a significant difference across conditions ($\chi2 = 12.9$, df = 3, $p = 0.004$). Results showed that visual assessment was not sufficient to complete the curvature task when the curve differences were smaller than $15 \pm 5sd$ mm. This visual assessment was part of the control condition introduced to make sure that the evaluation in our experiments with the shape display was based solely on the haptic experience delivered by the shape display and not by looking at the shape through the virtual reality system. To that end, we introduced Gaussian noise on the texture of the shape in virtual reality as well as made sure the edges were not visible.

## Threshold of Perception

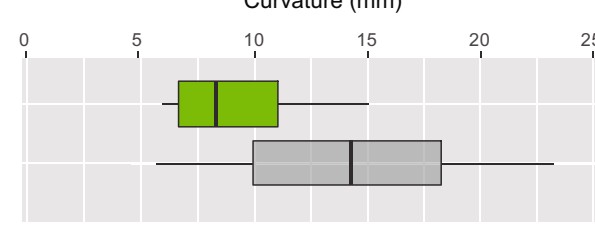

## Confidence

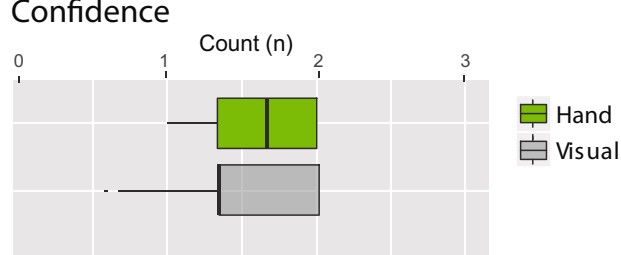

**Fig. 7 Results of experiment 2.** The threshold of perception ($n = 17$) is the point at which the answers to which of the two shapes was curvier started to be random. And on the right, we see the confidence that participants had in their assessment. Maximal confidence and perception were achieved with their hand touch. Paired Friedman significant difference across conditions for a threshold of perception ($\chi2 = 12.9$, df = 3, $p = 0.004$), participants reported significantly higher confidences when assessing by touching with their hand than visually ($p < 0.015$). For the boxplot: the center line represents the median; box limits represent upper and lower quartiles; whiskers the 1.5x interquartile range; and individual points are the outliers.

All in all, the visual assessment performance was far from the results achieved when participants were allowed to explore the curvatures with their Hand ($9 \pm 2sd$ mm), (Paired-Conover test with Bonferroni adjustment $p = 0.002$). Furthermore, participants reported significantly higher confidences when assessing with their hands than visually ($p < 0.015$, Figure 9).

Since participants did not reach the 5.5 mm vs. 5 mm differentiation we can say that our display had enough accuracy of rendering curvature to go beyond human sensing.

We believe also that the rigid but auxetic quality of the material is partially the reason why confidence in curvature assessment (Experiment 2) was higher for hand explorations.

These results validate the ability of the shape display as a proxy for the haptic perception of shapes of different curvatures, particularly if the shapes are presenting convexity.

**Experiment 3.** We recruited an additional 9 participants for a shape-moving experiment, in which the shape display can be moved freely on an area of 45 cm × 55 cm on top of a table (Fig. 8). We use sliders under the shape display platform to minimize friction so simply by pushing on the top of the shape display with the sensing hand participants can move and explore the space, as if they were using a mouse. In the same way, as in Experiment 1, participants were asked to choose which of the four shapes they were touching: a convex spherical surface, a concave spherical surface, a convex cylindrical surface (1D curve), and a concave cylindrical surface (Fig. 6). We also tested for the same three curvatures as in Experiment 1 (of 5, 10, and 25 mm).

Participants completed each shape (x4) in the different curvatures (x3) 3 times (x3). For every trial participants had to select which of the four possible shapes was the one they touched.

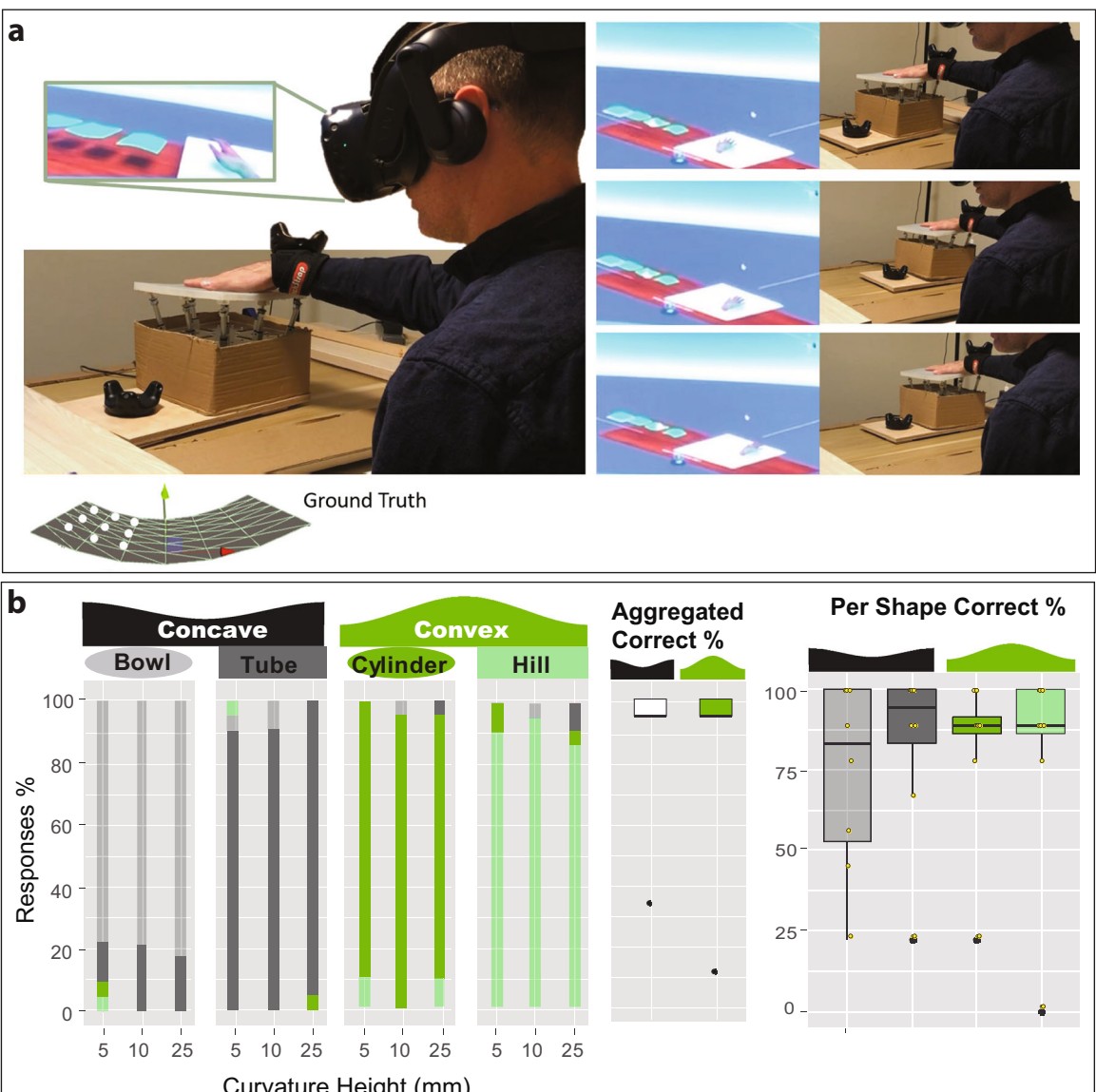

**Fig. 8 Moving shape in experiment 3. a** A participant exploring a large Concave cylinder used their hand to touch the surface and move the shape display around a larger space while in virtual reality. In this image, we can see a sequence of a participant exploring different areas of the large surface. **b** The results of the experiment ($n = 9$). In this graph, we present the histogram of classification responses of participants when presented with a particular shape in different curvatures. The boxplot shows the overall ability of the shape display to render concave or convex shapes of different types. For the boxplot: the center line represents the median; box limits represent upper and lower quartiles; whiskers the 1.5× interquartile range; and individual points are the outliers. Yellow dots represent individual participants. Participants had an overall accuracy of 80% in correct recognition of the shapes. No significant differences were found when the surface-displayed had different curvatures 25 mm, 10 mm, 5 mm, (paired Friedman analysis $\chi^2 = 1.6$, df = 3, $p = 0.63$). The recognition of surfaces was also not significantly different across different shapes (Convex or Concave, Conover test with Bonferroni adjustment $p > 0.094$).

The results (Fig. 8) showed that participants had an overall accuracy of 80% in correct recognition of the shapes overall. No significant differences were found when the surface displayed had different curvatures 25 mm, 10 mm, 5 mm, (paired Friedman analysis $\chi^2 = 1.6$, df = 3, $p = 0.63$). The recognition of surfaces was also not significantly different across different shapes (Convex or Concave, Conover test with Bonferroni adjustment $p > 0.094$). These results validate the ability of the shape display as a proxy for the haptic perception of shapes of different curvatures on larger surfaces in a dynamic way. They also show that the use of the shape display as a moving platform outperformed the results on Experiment 1 when the platform was static. This would

be aligned with prior research on the importance of propriceptive feedback loops to explore perception in virtual reality[23].

## Discussion

In our experimentation, we show that our prototype provides surface orientation, curvature, and shape information and that users are capable of discriminating curvature differences smaller than 5 millimeters. The material allowed the display to bend freely and generate shapes when we applied the actuators; and at the same time was able to resist local pressure up to 50N (as generated by touch) and maintain its shape, with a deformation

below 0.8cm. For example, when the shape display is moved around by the user sliding a mobile platform (Fig. 4).

The main difference to prior work is that there is no aliasing such as joints or steps between components of the display. Thus, a user can move their hand over the surface smoothly. There is some difference in resistance between the mounting point and the interpolating surface, but it is relatively small (see methods section for the results of the stress test). The most similar devices to ours are 1D, generally not motorized, and certainly not auxetic: they manipulate thin strips of material to construct a thin strip across a surface[32], or manipulate a developable surface (i.e. only zero Gaussian curvature as achievable by bending a flexible sheet into a section of a cylinder or cone)[33]. Therefore, mechatronic auxetic displays can be considered a separate class of haptic displays.

The layer of structured material might also become a useful complement not only to other types of displays but also for the larger set of soft robotic applications that require a minimal skeletal structure for the soft parts. In the field of haptics and shape displays we can envision multiple mergers, e.g, intermediate haptic displays that use the structured auxetic material together with more pins in a hybrid pin display with an auxetic surface. This could create both the low-frequency shapes and the high-frequency textures at the same time. Another alternative to provide texture frequencies on top of the material would be using voice coil actuators in a similar way that TORC has proposed which could also elicit illusions of elasticity[34] and compliance if attached to an array of force sensors on the surface of the material[35]. The use of a continuous surface offers easier skinning of further elements. Different types of force sensors or capacitive sensors could be added. We note that the use of a continuous, non-hinged, or jointed surface, simplifies the electrical connectivity issues that some shape-changing surfaces have. Our type of auxetic shape also has application in robotics and mechatronics to build control surfaces for rolling objects or controlling the flow of liquids or powders.

Through our experiments, we demonstrate the performance of our shape display in a series of virtual reality simulations (Figs. 6, 7, and 8). Current consumer virtual reality systems present visual and aural stimuli at high fidelity. While long considered a goal of virtual reality[36], the presentation of force and tactile (haptic) feedback lags behind[37]. Typical approaches to the presentation of force include grounded robots that the user can interact with. For example, the Haption Virtuose 6D device presents a handle that the user grips in a power grasp[38], where that handle is controlled by a linkage that can move the handle in six degrees of movement. Alternative approaches are hand-mounted robots and controllers[39]. These can present resistance or actuation of the fingers[40] but are often not grounded. A final class of approach is encounter-style haptics, where the user reaches out to touch an object or surface that moves or changes shape hidden from the user[41,42]. Our shape display can be considered encounter-style haptics that is grounded. Smaller versions could be mounted on controllers[11].

Some challenges remain in the area of shape displays such as the miniaturization of elements to produce smaller devices or represent higher spatial frequencies. Future work should also focus on the scalability and manufacturability of this type of shape display, including using 3D printing approaches.

In conclusion, we present a class of shape displays that can present large, curved surfaces using relatively small numbers of actuators and an auxetic surface. While the surface interpolates a rigid-to-the-touch surface between actuators, the mobility of the display and dynamic control of the actuators has shown high performance to render smooth large surfaces over the whole palm area at once, minimizing aliasing and enabling a powerful illusion of continuous surface exploration.

## Methods

**Driving & simulation.** The surface is moved with nine Actuonix L12-50-210-6-R linear actuators, with an extension range of 50 mm, a maximal speed of 6.8 mm/s, and a maximum force of 80 N. These actuators are not easy to back drive when static, so the length of the actuator remains fixed when the user pushes on the surface. Given the frame structure, the control points are effectively rigid with a minor exception: while the central actuator is fixed to move in a strictly vertical direction, each of the eight neighboring actuators is free to rotate along a plane defined by its bottom hinge and the middle actuator axis (Fig. 4). These planes all pass through a vertical line through the axis of the center actuator and are rotationally symmetric. The actuators pass through slots in the assembly constraining their motion to their corresponding plane. Through experimentation with a virtual prototype it was decided to set the base of the actuators narrower than the spacing of the actuator connectors on the surface. This allows the angle between the actuator and surface to be closer to orthogonal over a larger range of shapes.

The actuators were connected to the base using brackets. The connection to the surface uses ball and socket joints, constructed from layers of polycarbonate glued to the surface to have the rotation point as close to the top of the surface as possible (Fig. 4).

The simulation and graphics rendering of a 3D surface matching the physical display, along with the driver for the physical surface were developed in the Unity 3D software, version 2018.2.18f1. The simulation for the surface is based on a geometric proxy that mirrors the physical structure.

A simplifying assumption for simulation deforms the surface along radial lines from the assembly center to the actuators. The control system starts with a $3 \times 3$ array of target heights of the surface points. In a virtual prototype, these heights are controlled by blending target sets of heights that represent simple shapes, such as bowls, hills, saddles, and cylinders. We can then proceed in three steps to find the target actuator lengths. First, as the central actuator is rigid and only moves vertically, the actuator length is simply the target height. Second, we can set the four edge actuators by the construction in Fig. 4 where:

$$X = \sqrt{H^2 - (C - T)^2} \tag{1}$$

$$E = \sqrt{(X - A)^2 + T^2} \tag{2}$$

where $C$ = length of centre actuator (in mm determined prior) $H$ = spacing of edge hinge from centre across a surface (fixed at 90 mm) $A$ = spacing of actuator from centre across based (fixed at 60 mm) $T$ = target height (in mm) $E$ = edge actuator length to be determined (in mm)

Finally, we can set the four corner actuators by a similar method, where H and A are now multiplied by $\sqrt{2}$.

The linear actuators have a 50 mm range. With the length of the coupling on the surface, this gives the range from 130–180 mm. The actuators are controlled by a 12 channel Pololu Mini Maestro board connected by USB to a host computer. Pololu provides a dynamic link library (DLL) driver for the board, so integration into Unity was a straightforward import of this DLL. Within Unity, control is split into four scripts: DisplayDriver (drives the Pololu through the DLL), Shape Display (given a set of target heights and mechanical configuration of the display, calculate the actuator lengths), Shape Control (optional: provides a test interface that uses sliders on a window to control the shape), Shape Visualizer (optional: displays the actuators and also visualizes stretch or compression of mesh edges). The Shape Display script uses an external library to simulate the surface as a thin plate spline[43]. The library takes the rest positions of the vertices of a fine-scale mesh of the surface, the nine locations of the control points for the mesh (endpoints of the actuator hinges on the surface), and then calculates the shape of the fine-scale mesh and additionally the bending energy of the surface.

**Virtual reality system.** To support a virtual reality demonstration, we integrated the HTC Vive trackers and head-mounted display into the system. These were supported natively in Unity 2018. The standard HTC Vive setup includes two handheld controllers. We did not use these as we wanted to have the users explore with an unencumbered display. Instead, we attached a tracker to the hand of the participant, and another to the shape display as shown in Fig. 8. The users would see the surface as calculated by the Shape Display script based on the thin plate model. Given the HTC Vive coordinate system, we calculated a registration between the visual model and physical model, by measuring four points on the surface of the display with a controller together with the four corresponding virtual points on the surface. From this, a rigid affine transformation can be calculated to register the visual shape to the physical shape.

Despite the size of the display, the mechanism to control a surface lies behind it which can make the actuation completely hidden while operating. With virtual reality we have the distinct advantage that the user's vision is obscured by a head-mounted display, so the presentation of haptics can be achieved nicely with our device. We note that this might also be true for some situations in augmented reality, where the virtual object rendering totally occludes the surface and hence the mechanics of the device.

**Experiments**. To evaluate the resolution of our surface curvature interpolation between actuated positions we carried a set of user studies. We recruited 17 participants for the non-moving experiments 1 and 2, (aged $M = 38.8$ $SD = 7.5$, 4 female) and 9 additional participants for the moving experiment 3 (aged $M = 35$ $SD = 10.25$, 5 female). Our display is then tested by examining the ability of a user to explore and perceive the curvature of the object[44] through the different experiments. Since our display doesn't show high-frequency displacements or abrupt changes in curvature, we hypothesize very fine curvature changes will not so easily be found and detected by users as our surface operates at the threshold of human curvature perception.

Additionally, the user tests will show that the current display can render different types of surfaces with very fine differences. Our tests thus have three main thrusts: showing that bi-directional curvature is detectable for small amounts of curvature; reproducing the effect of using multiple fingers to detect curvature; and showing the feasibility of this device to render large surfaces.

Microsoft Research IRB approved the experimental protocol employed in the present study, and the experimental data were collected with the approval and written consent of each participant in accordance with the Declaration of Helsinki. The authors affirm that human research participants provided informed consent for publication of the images in Fig. 8.

## Data availability

The data diagrams supporting the manufacturing of the shape display generated in this study are provided in the Supplementary Information. The data that support the findings of the study are available in the Supplementary Information. The data diagrams supporting the manufacturing of the shape display generated in this study are provided in the Supplementary Information.

## Code availability

No specific firmware is required to operate the shape display. The servo driver is a Pololu Mini Maestro. That needs Windows driver software: https://www.pololu.com/docs/0J40/3.a which can be then driven through any project via loading the DLL.

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

## Author contributions

A. S. came up with the initial idea. All authors worked towards the use cases and implementation. A. S. manufactured the shape display with M. S. who helped with mechanical implementation and hardware choices. M. S. did the stress tests. E. O. did the curvature ground truth measurements with depth sensors and worked on the literature review and drawings. M. G. F., A. S. designed the experiments. M. G. F. and A. S. did the data analysis. All authors contributed on the writing of the paper.

## Competing interests

All authors were employed by Microsoft, an entity with a financial interest in the subject matter or materials discussed in this manuscript. Nonetheless, the authors declare that the current manuscript presents balanced and unbiased results, the studies were conducted following scientific research standards. Microsoft Research approved the experimental protocol employed in the present study, and the experimental data were collected with the approval and written consent of each participant in accordance with the Declaration of Helsinki.
