## [Peer Review File · Nature Communications]

Reviewers' comments:

Reviewer #2 (Remarks to the Author):

This paper presents a mechatronic system that renders surfaces with double curvature, which can support a user exploring a scene in virtual reality. The main contribution is the artifact being presented and initial user studies to demonstrate the usefulness in mimicking virtual surfaces.

In regards to the novelty of the system, there is an additional related paper that should be included to further position the present paper. Raun and Kirkegaard demonstrated a similar system to function as a reusable mould for thin shell concrete panels. While that system was not positioned as a haptic interface, the mechanical functions are quite similar. The present paper extends that work by reducing the number of actuators and by using an auxetic layer for structure.

Raun, C., & Kirkegaard, P. H. (2015, April). Adaptive Mould—A Cost-Effective Mould System Linking Design and Manufacturing of Double-Curved GFRC Panels. In 17th international congress of GRCA-GRC, Dubai.

While the system does represent the surfaces encountered in the virtual world. It would be very helpful for the authors to take a critical stance and further qualify the user experience. The video was very helpful to gain an understanding about how the movement of the device results in the surface changes. Figure 2 (right) and corresponding text did not explain that the user does not lift the hand from the surface, but will feel the transition as it is moved along a table. It is unclear what the users think and how they respond to the sensation and realism of the representation as the study focused on the accuracy of curvature perception. When a very large object is rendered with this device, the user would presumably lift the hand and place it to feel the surface. If the user kept their hand on the surface continually, the device does not provide the sheer forces and friction of the virtual surface. Do users adapt to this and do they remark about how this is different from real surfaces?

The user studies are overly summarized and it is unclear how the results generalize beyond the small sample size. What were the ages, gender, cultural background, experience with encountered haptics, etc? Were the same participants used in study 1 and 2? These details would be very helpful in evaluating this paper and in planning for future work that might extend this work.

The paper ends without a sharp conclusion. It would be a good opportunity to help the reader understand how to generalize this work or to set sights on the expected future work.

The text has many typographical errors to correct and some grammar corrections to be made. The following are some of these to illustrate:

Lines 69 and 156, "auxtetic" should be "auxetic"

Line 129 is the caption text for Figure 4 and it also self refers to Figure 4. Should it be Figure 3?

Lines 132-135 This sentence is very confusing and is unclear what is trying to be said. "up to 0.5 millimeters"? Perhaps "discriminate differences" should be "discriminating differences"?

Line 227 Error with reference, not sure which this is.
Line 213 "Polycarbonate" does not need to be capitalized
Line 269 "surface that is moves" should be adjusted for grammar
Line 275 "Figure 66" should be likely Figure 6
Line 298 "perceive curvature of object" should be adjusted for grammar
Line 299 "hypothesize a very fine curvature changes" should be adjusted for grammar
Line 342 Figure error, presumably Figure 8

Reviewer #3 (Remarks to the Author):

The authors describe a nice advance over conventional shape displays by the utilization of auxetic materials. There are elements of novelty in the paper but major issues need to be addressed.

a) The paper does not provide enough detail that it can be reproduced. Dimensions and all materials and processes need to be provided in detail in an academic paper so that anyone in the world can reproduce it, which is not possible here. Moreover some figures are hard to follow such as the designs etc.

b) A major challenge of the approach is that manufacturing and / or miniaturization of the elements would be very challenging. If the authors could demonstrate 3D printing or other approach to scale up / manufacture then it would make the paper more compelling. In the absence of this, a discussion is needed on scalability and manufacturability.

Response to reviewers' comments in blue:

Reviewer #2 (Remarks to the Author):

This paper presents a mechatronic system that renders surfaces with double curvature, which can support a user exploring a scene in virtual reality. The main contribution is the artifact being presented and initial user studies to demonstrate the usefulness in mimicking virtual surfaces.

R2: In regards to the novelty of the system, there is an additional related paper that should be included to further position the present paper. Raun and Kirkegaard demonstrated a similar system to function as a reusable mould for thin shell concrete panels. While that system was not positioned as a haptic interface, the mechanical functions are quite similar. The present paper extends that work by reducing the number of actuators and by using an auxetic layer for structure.

Raun, C., & Kirkegaard, P. H. (2015, April). Adaptive Mould—A Cost-Effective Mould System Linking Design and Manufacturing of Double-Curved GFRC Panels. In 17th international congress of GRCA-GRC, Dubai.

We thank the reviewer for referring us to this paper. The visual shape of their conception does visually resemble to our system, but it is essentially different in its interactivity, fabrication and form factor.

This paper described a constructed mold for cement surface generation. It is not an interactive system, in fact it isn't meant for any motion: interaction, movement, haptics or automation. It is a manual construction, that may be built according to the architect use, and the workers can solve problems manually. For example, there was no need to deal with stretching of the approximating surface while maintaining consistent haptic feedback. Also their design has a much milder curvatures and much smaller dynamic range relative to the need of a haptic interaction device, lower force on actuators, or consistency as the device move and represents different part of a larger virtual surface.

We have added this reference and discussed it. (line 62)

We have also added a reference to

Yasaman Tahouni, Isabel P. S. Qamar, and Stefanie Mueller. 2020. NURBSforms: A Modular Shape-Changing Interface for Prototyping Curved Surfaces. In Proceedings of the Fourteenth International Conference on Tangible, Embedded, and Embodied Interaction (TEI '20). Association for Computing Machinery, New York, NY, USA, 403–409. DOI:<https://doi.org/10.1145/3374920.3374927>

which has been published since our last version of the manuscript. But does not provide the stability and controlled deformation of our solution.

R2: (addition to Figure 1 and line 50).

R2: While the system does represent the surfaces encountered in the virtual world. It would be very helpful for the authors to take a critical stance and further qualify the user experience. The video was very helpful to gain an understanding about how the movement of the device results in the surface changes. Figure 2 (right) and corresponding text did not explain that the user does not lift the hand from the surface, but will feel the transition as it is moved along a table. It is unclear what the users think and how they respond to the sensation and realism of the representation as the study focused on the accuracy of curvature perception. When a very large object is rendered with this device, the user would presumably lift the hand and place it to feel the surface. If the user kept their hand on the surface continually, the device does not provide the sheer forces and friction of the virtual surface. Do users adapt to this and do they remark about how this is different from real surfaces?

The reviewer is right, while the haptic display represents the height of the virtual surface, and prevent the user's hand from penetrating into the virtual object's volume, it does not render forces that are tangential to the surface such as friction between the shape display and the skin.

While the first study that looks at the rendering quality of the display is unaffected by this limitation of our display, we have carried out an additional study where the shape display has been used to render a larger object. Hence the shape display is moved around the space and we can better study the effect of friction, or the lack of it on the performance. The user slides the shape display on a table, as a mouse, and we were able to provide both the visual sense that shows the hand position in relation to the surface, as well as the proprioception of the user's arm support with the feeling of the movement of the user's palm relative to the virtual surface. The combination of both are stronger than the expectation of the skin shear force. It may very well, that today users are already tuned to lack of this friction when using a mouse. Indeed, our results show that our platform performed even better in larger objects with a moving display than on the smaller not moving shapes.

This discussion has been added as part of the additional experiment. (lines 468 – 503, and figure 11 and 12)

R2: The user studies are overly summarized and it is unclear how the results generalize beyond the small sample size. What were the ages, gender, cultural background, experience with encountered haptics, etc? Were the same participants used in study 1 and 2? These details would be very helpful in evaluating this paper and in planning for future work that might extend this work.

We have added more information on the data captured, demographics and other relevant information about the participants. We had reduced it to fit on the letter format and space restrictions. But we have now resurfaced to the main manuscript. (lines 313-315)

The paper ends without a sharp conclusion. It would be a good opportunity to help the reader understand how to generalize this work or to set sights on the expected future work.

We have added a new paragraph to highlight the conclusions and way forward. We hope that it will be a much better experience for the readers. (lines 179-188)

The text has many typographical errors to correct and some grammar corrections to be made. The following are some of these to illustrate:

Lines 69 and 156, "auxtetic" should be "auxetic"

Line 129 is the caption text for Figure 4 and it also self refers to Figure 4. Should it be Figure 3?

Lines 132-135 This sentence is very confusing and is unclear what is trying to be said. "up to 0.5 millimeters"? Perhaps "discriminate differences" should be "discriminating differences"?

Line 227 Error with reference, not sure which this is.

Line 213 "Polycarbonate" does not need to be capitalized

Line 269 "surface that is moves" should be adjusted for grammar

Line 275 "Figure 66" should be likely Figure 6

Line 298 "perceive curvature of object" should be adjusted for grammar

Line 299 "hypothesize a very fine curvature changes" should be adjusted for grammar

Line 342 Figure error, presumably Figure 8

We thank the reviewer for their notes of typos and grammar errors and we did our best to correct them in the new text.

Reviewer #3 (Remarks to the Author):

The authors describe a nice advance over conventional shape displays by the utilization of auxetic materials. There are elements of novelty in the paper but major issues need to be addressed.

We are pleased to see the reviewer was aligned in the novelty of our approach. We have tried to address each of their issues for the manuscript.

a) The paper does not provide enough detail that it can be reproduced. Dimensions and all materials and processes need to be provided in detail in an academic paper so that anyone in the world can reproduce it, which is not possible here. Moreover some figures are hard to follow such as the designs etc.

We have added all the fabrication cads to the supplementary materials to make sure it is easy to reproduce.

b) A major challenge of the approach is that manufacturing and / or miniaturization of the elements would be very challenging. If the authors could demonstrate 3D printing or other approach to scale up / manufacture then it would make the paper more compelling. In the absence of this, a discussion is needed on scalability and manufacturability.

We have added these challenging areas to the manuscript, including also future work focus. (lines 179-188)

** See Nature Research's author and referees' website at www.nature.com/authors for information about policies, services and author benefits

REVIEWERS' COMMENTS

Reviewer #2 (Remarks to the Author):

In the revised article, the authors have improved the work and strengthened the focus on the user experience of the system and additional thoughts about how the system applies to future work and other research areas. The authors added additional details in relation to the user evaluations, related work, and have addressed the main concerns from the previous review.

Some additional quick comments for this current version:

In the rebuttal/revised work, there is a claim that the conclusion and thoughts about future work are included, which they are and has been addressed well. The line numbers in the rebuttal noted L179-188, however, I assume the authors meant to identify other line numbers, L170-179.

A small typo:

L170 "Sill some challenges remain" , should be "Still..."

Signed (Timothy Merritt)

Point by point response to the reviewers

REVIEWERS' COMMENTS

Reviewer #2 (Remarks to the Author):

In the revised article, the authors have improved the work and strengthened the focus on the user experience of the system and additional thoughts about how the system applies to future work and other research areas. The authors added additional details in relation to the user evaluations, related work, and have addressed the main concerns from the previous review.

Some additional quick comments for this current version:

In the rebuttal/revised work, there is a claim that the conclusion and thoughts about future work are included, which they are and has been addressed well. The line numbers in the rebuttal noted L179-188, however, I assume the authors meant to identify other line numbers, L170-179.

The reviewer is correct we referred to the wrong line numbers as the reviewer correctly interpreted. This discrepancy was there depending if the reviewer checked the version with tracked changes or the version without tracked changes.

A small typo:

L170 "Sill some challenges remain" , should be "Still..."

Thank you for catching this up. We have fixed this typo